# Transcriptional Regulation of Ripening in Chili Pepper Fruits (*Capsicum* spp.)

**DOI:** 10.3390/ijms222212151

**Published:** 2021-11-10

**Authors:** Maria Guadalupe Villa-Rivera, Neftalí Ochoa-Alejo

**Affiliations:** Departamento de Ingeniería Genética, Unidad Irapuato, Centro de Investigación y de Estudios Avanzados del Instituto Politécnico Nacional, Irapuato 36824, Mexico; gvillarivera@gmail.com

**Keywords:** ripening, transcription factors, chili pepper, capsaicinoids, carotenoids, anthocyanin, ascorbic acid

## Abstract

Chili peppers represent a very important horticultural crop that is cultivated and commercialized worldwide. The ripening process makes the fruit palatable, desirable, and attractive, thus increasing its quality and nutritional value. This process includes visual changes, such as fruit coloration, flavor, aroma, and texture. Fruit ripening involves a sequence of physiological, biochemical, and molecular changes that must be finely regulated at the transcriptional level. In this review, we integrate current knowledge about the transcription factors involved in the regulation of different stages of the chili pepper ripening process.

## 1. Introduction

The chili pepper belongs to the genus *Capsicum* and the Solanaceae family. The genus *Capsicum* encompasses 38 species, five of which have been domesticated: *C. annuum*, *C. frutescens*, *C. chinense*, *C. baccatum* and *C. pubescens* [1,2]. These species present different growth forms and a wide variety of shapes and colors of corollas and fruits [3]. Fresh or dry chili pepper fruits have economic and culinary importance, and they are cultivated and distributed worldwide; moreover, these fruits have generated growing interest because of their versatility and innumerable uses as condiments and medicinal properties [4,5]. Chili pepper fruits are commonly classified as berries, which are indehiscent fruits with many seeds bound to the pericarp. *Capsicum* fruits present variable shapes (spherical, conical, ovoid, fusiform, or elongate), colors (red, orange, yellow, white, purple or brown) and pungent and nonpungent varieties [3,4]. Traditionally, *Capsicum* fruits have been used for their flavor, spice, and color, and applied to cure toothache, muscle pain, neuralgia, rheumatism, and diabetes. Additionally, extracts of chili pepper have been investigated as antimicrobial agents against fungi and viruses and as insecticidal, anthelmintic and larvicidal agents; moreover, the anti-inflammatory properties, cardiovascular protection effects, anti-obesity effects, antioxidant activities, anticancer activities, antiangiogenic and anti-neoplastic effects, and antithrombotic and vasodilatory properties of chili pepper extracts have been widely described [6]. The nutraceutical effects of extracts of chili pepper fruits are due to the diverse bioactive compounds located in the placenta, pericarp and seeds [7], and these bioactive metabolites consist of capsaicinoids, carotenoids, phenolic compounds (such as flavonoids) and volatile compounds [8,9]. In addition, *Capsicum* fruits have high levels of vitamin C, pro vitamin A, and minerals, such as calcium and iron, among others, all of which have health-promoting potential. Furthermore, chili pepper phytochemicals constitute natural compounds for the cosmetic, pharmaceutical, and agro-food industries, where they are used as substitutes for synthetic ingredients [10,11].

The maturation of fruits in plants involves different steps: fruit set, fruit development, and fruit ripening. Fruit ripening requires a sequence of biochemical and physiological transformations (modifications of the plant cell wall, conversion of starch to sugars, alterations in pigment biosynthesis, accumulation of aromatic volatile compounds, among others) that are regulated genetically and take place at the final stage of fruit development [12,13]. This process includes visual changes, such as fruit coloration, and changes in flavor, aroma, texture and palatability, which make the fruit edible, attractive and desirable for consumers and increase its quality and nutritional value [14]. According to physiological variations in the pattern of respiration during the ripening process, fruits are classified as climacteric (increase in levels of CO_2_ and ethylene production detected) and non-climacteric (no peak in ethylene emission detected). For example, tomato, apple, and avocado are classified as climacteric fruits, while lemon, pineapple and chili pepper represent important models for the analysis of non-climacteric fruits [15]. During the ripening process of chili pepper fruits (Figure 1), gene expression, metabolite accumulation and antioxidant activity change through the different stages of growth and maturation [16]. These changes must be highly controlled, and transcription factors (TFs) are crucial components for regulating the expression of genes involved in the molecular mechanisms that occur in this process [12]. Although transcriptional regulation of the ripening process in tomato (the closest relative to *Capsicum*) has been widely studied [17], both species likely have different ripening mechanisms and therefore different regulatory effects on gene expression [18]. Transcriptional regulation is extremely important during the ripening process, and during the ripening process between tomato and chili pepper fruits, differences are observed in the transcription of genes encoding TFs, suggesting the existence of key factors that differentiate the ripening processes in climacteric and non-climacteric fruits [19]. In this review, we integrated current knowledge about the transcription factors involved in chili pepper fruit development and incorporated available information about the transcriptional regulation of capsaicinoid, carotenoid, anthocyanin and ascorbic acid biosynthetic pathways as ripening-related processes.

## 2. Chili Pepper Ripening Regulation Process

Metabolic changes that occur during the ripening process of chili peppers cause transformations in flavor, color, texture and aroma. Accordingly, different transcription profiles of genes that encode key enzymes of metabolite biosynthetic pathways and distinct antioxidant activity patterns during chili pepper ripening have been described at the early (16–25 DPA), breaker (36–28 DPA) and later stages (43–48 DPA). For instance, the production of glycosides (quercetin, luteolin and apigenin) characterized the early stages, while the production of other metabolites, such as shikimic acid, putrescin and γ-aminobutyric acid (GABA), which were highly accumulated, gradually decreased at the breaker stage. Finally, the content of amino acids (leucine, isoleucine, proline and phenylalanine) and other compounds, such as kaempferol, capsaicin and dihydrocapsaicin, increased significantly at the later stage [16]. Transcriptomic analyses conducted in two stages of ripening (immature (green fruit) and mature (red fruit)) of chili pepper fruits revealed 1008 differentially expressed genes in these fruits. Distinct expression patterns are associated with different metabolic pathways, such as carotenoid and phenylpropanoid biosynthesis, plant hormone signal transduction, secondary metabolites and sesquiterpenoid and triterpenoid biosynthesis pathways. Additionally, a comparison between the transcription patterns of *C. annuum* L. (‘Tampiqueño 74’, a domesticated variety) and *C. annuum* L. var. *glabriusculum* (chiltepin wild variety) resulted in 53 differentially expressed genes associated with size, shape and secondary metabolite biosynthesis, suggesting that these differences could be related to the domestication process [20].

Although chili pepper has been used as a model for the study of non-climacteric fruits, variations in the respiration rates and ethylene production among cultivars of *Capsicum* have been detected because of the diversity of this genus [21]. Accordingly, genes encoding key enzymes of ethylene biosynthesis, such as 1-aminoacyl cyclopropane 1-carboxilate (ACC) synthase (ACS) and ACC oxidase (ACO), were not expressed in chili pepper fruits [19]. Nevertheless, a proteomic analysis conducted with the aim of identifying the differentially expressed proteins in three ripening stages (green, breaker red and light red) of *C. annuum* L. demonstrated that at least one isoform of ACO (CaACO4) was expressed at the breaker stage of chili pepper ripening, and these results suggest that some aspects of regulation may be conserved in climacteric and non-climacteric fruits [22]. Other evaluations of the levels of transcription of genes encoding several isoforms of ACO, ACS and the ethylene receptor (ETR) demonstrated that except for *CaACO4*, the expression of these genes was limited to *Capsicum* fruits; additionally, the activity levels of ACS and ACC were very low, suggesting that ACS activity was the limiting step in the ethylene biosynthesis pathway in chili pepper, thus leading to the restriction of ACC accumulation [18]. Treatment with ethylene or 1-methylclopropene likely differentially regulated ethylene production in non-climacteric chili pepper fruit compared with climacteric tomato; moreover, the presence and upregulation of the *CaACO4* gene in *C. annuum* L. fruits suggests that ethylene-independent pathways may exist in non-climacteric fruits [18,23].

The transcripts of three genes encoding TFs related to ethylene-mediated signaling (*EIL1*-like) were detected in *C. annuum* at the brown stage [19]. EIL TFs are homologous to the ethylene insensitive (EIN) TFs reported in *Arabidopsis thaliana* and *Solanum lycopersicum*, and they act downstream from ethylene receptors in the ethylene signal transduction pathway and are described as positive regulators of ethylene responses [24]. These results might suggest that activation of ethylene signaling and the subsequent alteration of ethylene sensitivity could be conserved in non-climacteric fruits [18,19]. In contrast, upregulation of ethylene receptor-like protein (ETR), a key regulator of the ethylene transcription pathway, was detected in a transcriptomic analysis of *C. frutescens* and *C. annuum* performed at different stages of ripening, suggesting that at least some species of chili pepper exhibit climacteric behavior [20,25].

A comparative transcriptome analysis performed between tomato (climacteric) and chili pepper (non-climacteric) showed that the transcript profiles of genes encoding TFs such as the ripening inhibitor (RIN; a regulator or ripening in tomato) [26], tomato AGAMOUS-LIKE1 (TAG1; which has an important role in the regulation of both fleshy fruit expansion and ripening process) [27], and non-ripening (NOR) (Figure 2A) [12] are conserved in both types of fruits [18]. In contrast, other genes encoding TFs involved in the regulation of fruit ripening, such as *Colorless ripening* (*Cnr*; a major component in the regulatory network controlling tomato ripening) [28], *Uniform* (encodes a Golden 2-like (SlGLK2) transcription factor involved in chlorophyll accumulation and distribution during fruit development) [29], and *LeHB-1* (participates in the control of ripening) [30], were expressed at low levels during chili pepper ripening and highly expressed during the tomato ripening process [18].

Some transcription factors have been identified and proposed as regulators of the transcription of structural genes that encode proteins involved in the ripening process of chili pepper fruits. Among the differentially expressed genes (DEGs) identified in the transcriptomic analysis conducted in immature and mature chili pepper fruits, only four genes encoding TFs were upregulated (*FUL2*, two *bZIPs* and one *homeobox TF*), while 25 TF genes were downregulated [20].

With the aim of evaluating the role of some TFs during the chili pepper ripening process, functional and interaction assays were performed. The gene expression profile and yeast two-hybrid assays suggested that the interaction of two transcription factors (CaMADS, subfamily AGL2; and CAMADS6, subfamily SQUA) could have an important role in the fruit development of a hot chili pepper [31]. The MADS (MCM1 AG DEFA SRF) box is a conserved family of TFs involved in the control of development and signal transduction [32]. Structurally, MADS-box TFs bear two conserved domains: the MADS domain of 56 amino acids, which includes the motif of union to the consensus DNA sequence (Table 1) and the heterodimerization zone, and the K domain (required for MADS protein interaction), which plays an important role in MADS-box proteins. Moreover, the structure of these TFs includes two divergent regions: the interdomain region (I-region) located between the MADS and K domains and the C-region, which is located at the C-terminus [33,34,35,36]. With the aim of investigating the function of other MADS-box TFs from chili pepper, the role of CaMADS-ripening inhibitor (CaMADS-RIN), a TF belonging to the *SEPALLATA* (*SEP*) group, was evaluated. The *CaMADS-RIN* gene showed high transcription levels during the ripening process of *C. annuum*, whereas its heterologous expression in tomato plants revealed that this TF had a positive role in ethylene biosynthesis and fruit ripening in tomato, suggesting its involvement in the fruit ripening of either climacteric or non-climacteric fruits [37]. A comparative genomic analysis based on the tomato genome was carried out to identify genes involved in the development/ripening process of chili pepper fruits; in this case, 32 orthologous genes encoding proteins involved in cell wall formation, metabolite accumulation, coloring, softening and aroma production were expressed, and only 12 of them were differentially expressed in *Capsicum*. Interestingly, it was possible to identify a gene with high levels of expression at the breaker stage and mature fruits by quantitative expression analysis; this gene was identified as an ortholog gene of *MADS-RIN* by Dong in 2014 [37], which supported its possible participation in the fruit ripening process of *Capsicum* [25].

Moreover, a wide genomic and subcellular localization analysis of the auxin response factor (ARF) family from *C. annuum* L. identified *CaARF* genes that exhibited higher levels of expression in flowers (*CaARF18*) and fruits (*CaARF4* and *CaARF5*) than in other chili pepper organs [38]. ARFs are a family of TFs involved in the transcriptional regulation of genes of the auxin response, and they interact with the auxin response elements located in the promoters of genes via the conserved B3 DNA binding domain (DBD) of ARFs. In addition, a variable middle region (MR) that works as a repression or activation domain and a conserved C-terminal domain (CTD), which enables protein–protein interactions, compose the ARF structure [50,51]. In this sense, it is possible to hypothesize that *CaARF18*, *CaARF4* and *CaARF5* could participate in the regulation of chili pepper fruit development and ripening processes.

Other important candidates for chili pepper ripening regulation are WRKY TFs. *CaWRKY*s are expressed at higher levels in vegetative tissues, and a previous study indicated that almost 60% of these TFs were expressed during chili pepper fruit maturation and regulated by plant hormones and abiotic stress, suggesting their key regulatory role in the ripening process [52]. A highly conserved WRKY domain at the N-terminal region characterizes WRKY TFs. The WRKY domain contains a conserved peptide WRKYGQK and a zinc-finger structure (Cx4–5Cx22–23HxH or Cx7Cx23HxC) [53].

Finally, the role of noncoding RNAs (ncRNAs) was evaluated. The transcriptional profile of ncRNAs showed that they are differentially expressed in mature green and red ripe chili pepper fruits. Moreover, the targets of these ncRNAs were suggested to include TFs, such as ARF, bHLH, bZIP, ERF, MYB, NAC and WRKY, and enzymes involved in cell wall remodeling, pigment accumulation, fruit flavor and aroma development, and ethylene production; thus, these ncRNAs constitute a new level of control during the fruit ripening process [54].

## 3. Transcriptional Regulation of Capsaicinoids Biosynthesis

A characteristic quality of chili pepper fruits is pungency, and they produce and accumulate hot compounds called capsaicinoids, which are alkaloids found exclusively in *Capsicum* species [55]. The most abundant capsaicinoids in chili pepper fruits are capsaicin (trans-8-methyl-N-vanillyl-6-nonenamide), a derivative of homovallinic acid, and dihydrocapsaicin. Interestingly, capsaicin has become a very important bioactive compound as a cancer preventive agent and potentially as a treatment against various types of cancer [56]. Capsaicinoid accumulation in *Capsicum* fruits is genetically determined and depends on the developmental stage, and it occurs in the fruits from the early stages (ten days after pollination), starting at the instant that growth begins, reaches a maximum concentration at 33–40 dpa depending on the cultivar, and then decreases due to the activity of different peroxidases [57,58]. Biotic (wounding, attack by pathogens or insects) and abiotic factors (temperature, light, water, levels of CO_2_, altitude, and nutrient availability) influence the content and accumulation of capsaicinoids. Abiotic stresses stimulate a drastic change in the expression levels of genes involved in the production of capsaicinoids [59,60]. For instance, in hot chili pepper, the level of pungency is highly variable in response to temperature and the accumulation of capsaicinoids was reported to increase by exposure to water stress. Finally, mineral nutrition (mainly nitrogen and potassium) has been observed to have a key role in the metabolism of capsaicinoids [57]. Moreover, the differences in the levels of pungency in distinct species of *Capsicum* are partially dependent on the genotype. The absence of capsaicinoids in species or varieties of chili pepper is due to the presence of mutations in genes encoding key enzymes or TFs that regulate the capsaicinoid biosynthetic pathway or related pathways [59,60]. Finally, the content of capsaicinoids in chili pepper fruits depends on the type of tissue, with synthesis and accumulation of these compounds occurring in the epidermal cells of placental tissues and an ambiguous boundary observed in the placental septum and pericarp [61].

Capsaicinoids are produced by the convergence of two metabolic pathways: the phenylpropanoid and branched-chain fatty acid pathways [62]. Transcriptional regulation of both metabolic pathways, as well as amino acid precursor synthesis, results in determinants of capsaicinoid production and accumulation (Figure 2B) [60]. The complexity of the regulatory network of capsaicinoid biosynthesis was evidenced through an analysis of the expression profile of fruits of the Indian chili pepper *C. frutescens* cv. Guijianwang during different developmental stages, which found that at least 20 candidate enzyme-encoding genes could be related to capsaicinoid biosynthesis [63].

With the aim of elucidating the mechanisms of transcriptional regulation of capsaicinoid biosynthesis, research has evaluated the potential participation of the MYB TF family in the regulation of several secondary metabolism pathways (lignin, vitamin C, phenylpropanoid, capsaicinoid, and carotenoid biosynthesis) [47]. Structurally, MYB TFs are organized in two regions: an N-terminal conserved DNA-binding domain of 52 amino acids (R2R3-type domain in plants), and a variable modulator region located at the C-terminal region responsible for the regulatory activity [64,65]. MYB TFs are involved in several cellular processes, such as cellular and organ morphogenesis, primary and secondary metabolism, chloroplast development, cell cycle control, and biotic and abiotic stress responses [43,65].

The isolation and characterization of *CaMYB31*, an *R2R3-MYB* gene from *C. annuum* ‘Tampiqueño 74’, and a positive correlation was observed between the transcription profile of *CaMYB31* and structural genes encoding enzymes of capsaicinoid biosynthesis [44]. Additionally, silencing of *CaMYB31* resulted in a decrease in structural gene expression and capsaicinoid content. Remarkably, the expression levels of *CaMYB31* were affected by plant hormones, wounding, temperature, and light. These results together suggest that the CaMYB31 TF has an important role in the regulation of the capsaicinoid biosynthesis pathway [44]. Moreover, a homologous gene of *CaMYB31* in fruits of *C. annuum* with different degrees of pungency (pungent cultivar ‘Tean’, and nonpungent accession ‘YMC334’) was identified [66], and this homolog was described as a single recessive gene named *Pun3*. Interestingly, *Pun3* was only transcribed in the pungent cultivar ‘Tean’, and a sequence analysis detected a premature stop codon in *Pun3* from the nonpungent accession; consequently, genes encoding enzymes of the capsaicinoid biosynthetic pathway were significantly downregulated in ‘YCM334’ compared to the pungent chili pepper cultivar—these results confirm the role of MYB31 as a master regulator of capsaicinoid biosynthesis [66]. Accordingly, a genetic and functional analysis was performed in five *Capsicum* species, including *C. chinense* (with a high content of capsaicinoids), and it identified the locus *Cap1* (identical to *Pun3*), which encodes the MYB31 TF and contributes to the level of pungency. *MYB31* is specifically expressed in placental tissue and activates the transcription of capsaicinoid biosynthesis-related genes, resulting in genus-specialized metabolite production. Additionally, an increase in the expression of *MYB31* (from *C. chinense*) caused by natural variations in its promoter allowed for the binding of the transcriptional activator WRKY9 and triggered an increase in the expression of capsaicinoid biosynthesis-related genes (Table 1) [67,68].

In addition to CaMYB31, the jasmonate-inducible transcription factor CaMYB108 was described [47]. Virus-induced silencing (VIGS) of *CaMYB108* led to a reduction in the transcript levels of capsaicinoid biosynthetic genes and capsaicinoid content in *C. annuum* [45]. Moreover, a weighted gene coexpression network analysis (WGCNA) identified a module related to capsaicinoid biosynthesis within this module, and a MYB TF (CaMYB48) was described as a key regulator of capsaicinoid biosynthesis in chili pepper [46]. Finally, expression analyses of *CaR2R3-MYB* genes and capsaicinoid biosynthesis-related genes reported that at least six CaR2R3-MYBs may regulate capsaicinoid biosynthesis in chili pepper fruits [69]. Interestingly, expression analysis of the candidates mentioned above (*CaMIB31*, *CaMYB108* and *CaMYB48*) and the six CaR2R3-MYBs *CaMYB47*, *CaMYB64*, *CaMYB73*, *CaMYB74*, *CaMYB87* and *CaMYB92* showed that only *CaMYB31* had a positive correlation with the transcription patterns of genes related to the capsaicinoid biosynthesis pathway [47]. Additionally, coexpression analysis allowed us to identify three strong candidate genes for capsaicinoid biosynthesis regulation: *CaMYB103*, *CaMYB115* and *CaDIV14*, whose expression profiles correlated positively with those of *AT3* (encoding an acyl transferase; putatively capsaicinoid synthase) and *CaMYB31* [47]. In particular, *CaDIV14* was grouped with DIVARICATA-like genes, which encode a subgroup of MYB-related proteins involved in the determination of the dorsoventral asymmetry pattern in *Antirrhinum* flowers [70].

On the other hand, multiple members of the ethylene response factor (ERF) transcription factor family have been proposed as regulator candidates of genes encoding structural capsaicinoid biosynthesis-related enzymes. ERFs belong to the apetala2/ERF (AP2/ERF) superfamily of TFs. Members of the ERF family possess a single DNA binding domain AP2/ERF, which is composed of 60–70 amino acids; moreover, the induction of *ERF* genes is not always dependent on ethylene action, and these genes are involved in the regulation of several biological processes related to growth and development as well as in the responses to biotic and abiotic stresses [41,71,72]. An evaluation of the transcription profiles of two ERF TFs (*Erf* and *Jerf*) in nine cultivars of chili pepper with different capsaicinoid contents demonstrated that these genes were positively correlated with pungency. Maximum levels of expression of both genes were detected at 16–20 dpa prior to the accumulation of capsaicinoids in the placental tissues. Thus, these two ERF family members could participate in the regulation of the capsaicinoid biosynthetic pathway [73]. Subsequent analysis confirmed that the transcription patterns of three genes encoding TFs of the ERF family (*CaERF102*, *CaERF53*, *CaERF111* and *CaERF92*) were similar to the capsaicinoid accumulation pattern, and CaERF92, CaERF102 and CaERF111 may be involved in the regulation of capsaicinoid biosynthesis mediated by temperature [42].

Ultimately, TFs belonging to the basic/helix-loop-helix (bHLH) family have emerged as new regulator candidates for capsaicinoid accumulation in *Capsicum* placenta. This superfamily of proteins is characterized by the presence of the bHLH conserved domain of approximately 60 amino acids and two functionally distinctive regions: an N-terminal basic region of approximately 15 amino acids (mainly basic residues) involved in DNA binding, and a C-terminal HLH region that functions as a dimerization domain, which is composed of hydrophobic residues [74]. Accordingly, the expression levels of the *CabHLH007*, *CabHLH009*, *CabHLH026*, *CabHLH063* and *CabHLH086* genes were positively correlated with the profile of capsaicinoid accumulation [18]. Moreover, CabHLH007, CabHLH009, CabHLH026 and CabHLH086 could be involved in temperature-mediated capsaicinoid biosynthesis. Interestingly, yeast two-hybrid (Y2H) assays demonstrated that CabHLH007, CabHLH009, CabHLH026, CabHLH063 and CabHLH086 might interact with MYB31, the master regulator of capsaicinoid production [39].

## 4. Transcriptional Regulation of Carotenoid Biosynthesis

The different colors of chili pepper fruits are due to variations in carotenoid composition and content in the pericarp [10,75]. Carotenoids are lipophilic molecules chemically composed of 40 carbons with conjugated double bonds. According to their composition, they are classified as carotenes (containing C-H atoms) and xathophylls (containing C-H-O atoms) [76]. In plants, carotenoids have important functions because they constitute the major attractants for pollinators and enhance the flavor of food crops. Additionally, carotenoids offer photoprotection against photooxidative damage in plant cells, increase heat and light stress tolerance, are involved in photosystem assembly, represent precursors to phytohormones, such as abscisic acid (ABA) and strigolactons, and are rhizosphere signal molecules [77]. Similar to capsaicinoids, carotenoids have outstanding nutraceutical properties in the prevention and treatment of human diseases and present antioxidant, cardiovascular disorder preventive, anti-obesity, anti-inflammatory and cancer preventive activities [78].

The variable accumulation of carotenoids in the fruits of *Capsicum* spp. is determined genetically and depends primarily on the cultivar and the development and ripening stages. For instance, red chili pepper fruits present high contents of capsanthin, capsorubin, β-cryptoxanthin, β-carotene, zeaxanthin and antheraxanthin, whereas brown fruits accumulate the same pigments as red fruits as well as high amounts of chlorophyll B and lutein [79]. Yellow fruits contain high levels of violaxanthin and lutein and antheraxanthin in a minor proportion. Finally, the orange color of chili pepper fruits is conferred mainly by the accumulation of zeaxanthin and lutein [79,80].

In the initial stages of chili pepper fruit development, the characteristic immature green color is obviously due to the presence of chlorophyll. Chloroplast development, compartment size and chlorophyll content variations in immature chili pepper fruits have been associated with the allelic diversity of three genes: *C. annuum* Golden2-like (*CaGLK2*) [81], *C.*
*chinense* zinc finger LSD-One Like 1 (*CcLOL1*) and *CcAPRR2* [82]. Additionally, the color of mature chili pepper fruits is regulated by three independent loci: *C1*, *C2* (encoding phytoene synthase (PSY)) and *Y* (encoding capsanthin/capsorubin synthase (CCS)) [83]. A recent report demonstrated that locus *C1*, which controls the white fruit color, encodes a homolog of a pseudoresponse regulator 2-like (PRR2) of *A. thaliana*. Virus-induced gene silencing (VIGS) assays performed in *C. frutescens* plants with a *PRR2* construct produced a lighter fruit color [84]. Moreover, the overexpression of *PRR2* in tomato showed enhanced levels of chlorophyll in fruits at the early stages of development and a slight increase in the amount of carotenoids in red fruits due to the expansion of the plastid number [85]. Moreover, PRR2 transcription appears to be regulated by the circadian rhythm and could interact with the abscisic acid insensitive 3 (ABI3) protein [86].

In plant cells, carotenoids are stored in chromoplasts (plastids that have lost photosynthetic activity) [87]. During the ripening process, the characteristic grana-intergranal structure of chloroplasts is degraded and substituted by nonchlorophyll single thylakoids derived from the inner envelope membrane, and these morphological changes are correlated with metabolic transformations, such as the loss of galactolipids, a slight increase in phospholipid content, and the esterification of xanthophylls with fatty acids [88,89]. Chromoplasts are unique plastids capable of accumulating massive amounts of carotenoids, and these pigments are stored in membranes and fibrilar, crystal and tubule structures [87]. Particularly, plastoglobules (oval or tubular structures rich in lipids that are located in all plastids) constitute active metabolic sites of carotenoid biosynthesis and recycling. Proteomic analysis of chromoplast plastoglobules revealed the presence of seven fibrillins and 25 proteins involved in the metabolism of lipids, isoprenoid-derived molecules, and carotenoid conversion [90]. Interestingly, differences in the abundance of chromoplast proteins were observed in other plastid types and among different crops, and such differences have been observed for *Capsicum* spp., which accumulate high levels of CCS enzyme and fibrillins inside the chromoplasts [91,92].

Carotenoids in plants are biosynthesized through the methylerythritol phosphate-derived pathway [93]. With the exception of ζ-carotene isomerase (Z-ISO), most carotenogenic enzymes have been localized in the chloroplast proteomes of chili pepper fruits [92,94]; additionally, the transcription profiles of genes encoding carotenogenic enzymes during the chili pepper fruit development process have been reported [95]. Phytoene, the first colorless carotenoid, is synthetized in the stroma of chromoplasts, and the final desaturation and cyclization stages that produce colored carotenoids occur in the membrane [96]. The biosynthesis of phytoene, carotene and capsanthin depends on the phospholipid environment inside the chromoplast and the development of new membranes during the conversion of chloroplasts into chromoplasts [88].

In plants, the biosynthesis of carotenoids is regulated at different levels, such as metabolite feedback and transcriptional and epigenetic control [97]. Mechanisms of the transcriptional regulation of carotenoid biosynthesis are highly relevant and have been widely studied in climacteric fruits, with tomato (*Solanum lycopersicum*) used as a model [17]; nevertheless, our understanding of the transcriptional regulation of carotenoid biosynthesis remains fragmented and incomplete, especially in non-climacteric fruits (Figure 2C) [98].

Transcriptional regulation of the first step of the carotenoid biosynthetic pathway seems to be ubiquitous in plants, and phytochrome interacting factor 1 (PIF1) downregulates the content of carotenoids by repressing the expression of the gene encoding the phytoene synthase (PSY) enzyme, which is the main rate-limiting enzyme of the pathway [99]. PIF proteins belong to the bHLH family of TFs, and they are categorized as negative regulators (repressors) of chlorophyll production [100]. PIFs are capable of transducing light signals to control transcription and, consequently, carotenoid accumulation [99]. Additionally, it has been suggested that carotenoid biosynthesis is subjected to fine regulation integrated by specific modules of antagonistic TFs and cofactors. In this sense, the transcriptional cofactor phytochrome-rapidly regulated 1 has been proposed to prevent the union of TFs to their target promoters, and this cofactor interacts with the PIF1 repressor and promotes the transcription of the PSY enzyme-encoding gene [101].

Although the transcriptional regulation of genes encoding carotenogenic enzymes is poorly understood in *Capsicum* spp., transcriptomic and functional analyses have identified the role of some TFs that control this important process in non-climacteric fruits. Transcriptomic comparisons during the maturation stages of climacteric and non-climacteric fruits have revealed that although there is a common set of metabolic genes that controls the biosynthesis and accumulation of carotenoids, some specific regulators could differ between climacteric and non-climacteric fruits, which is probably because of their distinct use and susceptibility to ethylene during the ripening process [102].

An analysis of the ripening of chili pepper fruits (*C. frutescens*) showed that ethylene emissions increased at the initial coloring stage and reached a peak at the brown stage [103]. In contrast, the levels of ABA showed a slight increase before the brown stage and peaked at the full red stage. RNA sequencing (RNA-seq) and VIGS analysis showed the role of ACO3 and NCED1/3 during the ripening of *C. frutescens* fruits. Higher *ACO3* transcription is usually related to an ethylene emission peak, which positively regulates the change in fruit color through the carotenoid biosynthesis pathway, whereas higher *NCED1/3* expression is associated with ABA accumulation and subsequent degradation of chlorophyll during the ripening process [103].

Some transcriptional regulators of the carotenoid biosynthesis pathway have been described for non-climacteric fruits. The citrus gene *CrMB68*, which encodes an R2R3-MYB TF, showed direct and negative regulation of β-carotene hydroxylase (*CrBCH*) and 9-*cis*-epoxycarotenoid dioxygenase (*CrNCED*) gene expression [104]. On the other hand, overexpression of the *CsMADS6* gene in sweet orange (*Citrus sinensis*) triggered enhanced expression of the *CsPSY*, phytoene desaturase (*CsPDS*), carotene isomerase (*CsCRTISO*) and *CsBCH* genes, and inhibited the expression of the ε-lycopene cyclase (*CsLCYE*) gene. Additionally, a yeast one-hybrid assay established that the CsMADS6 TF interacts with the promoters of the β-lycopene cyclase 1 (*CsLCYB1*) and *CsLCYB2* genes [105].

Regarding the transcription factors that may regulate the expression of genes encoding carotenogenic enzymes in chili pepper fruits, the expression profile of the ERF TFs *CaERF82*, *CaERF97*, *CaERF66*, *CaERF107* and *CaERF101* showed a positive correlation with the levels of β-carotene, zeaxanthin and capsorubin, suggesting their possible regulatory role in the carotenoid biosynthetic pathway [42]. Related analyses conducted with the bHLH family of TFs from chili pepper demonstrated that the transcription profiles of the *CabHLH009*, *CabHLH032*, *CabHLH048*, *CabHLH095* and *CabHLH100* genes were similar to the carotenoid biosynthesis profile in the pericarp, including lutein, zeaxanthin and capsorubin (Table 1) [39].

Transcriptomic analysis and carotenoid content profiling were carried out in orange and red chili peppers at 25, 40 and 55 days DPA. Twenty-three DEGs involved in the carotenoid biosynthetic pathway were reported [49]; interestingly, significant differences were observed in the expression profiles of the β-carotene hydroxylase 1 (*CHYB1*) and *CCS* genes of red and orange fruits during growth and maturation. Differences in the regulation of the *CCS* gene likely produce distinct accumulation of carotenoids and fruit colors.

On the other hand, a positive correlation in the expression profile of the *CCS* gene and two genes encoding TFs belonging to the MYB family, *CADIV1* and *CaMYB3R-5*, has been reported [47], and these TFs constitute interesting candidates for future functional analysis.

Finally, a gene coexpression network analysis identified the following six TF candidates for *CCS* gene regulation: U-box domain-containing protein 52, GATA transcription factor 26, RING/FYVE/PHD-type, F-box protein SKIP23, CONSTANS-LIKE 9, and zinc finger family FYVE/PHD-type (Figure 2C, Table 1) [49]. However, additional experiments are required to confirm its regulatory role in the carotenoid biosynthetic pathway.

## 5. Transcriptional Regulation of the Anthocyanin Biosynthetic Pathway

Some species of chili pepper biosynthesize and accumulate anthocyanins in their tissues and organs (flowers, fruits and foliage), and these phenolic compounds are responsible for red, blue, and purple colors, with purple being the most frequent color observed in *Capsicum* spp. Anthocyanins are secondary metabolites derived from the phenylpropanoid pathway [106].

Because of their antioxidant properties, the effect of anthocyanins in the prevention of chronic cardiovascular diseases and diabetes has been studied, and their anticancer, anti-obesity, antimicrobial, cardiovascular disease prevention and visual health promotion activities have also been reported [107].

Anthocyanin accumulation is regulated by the interaction of MYB, MYC and WD40 TFs with promoters of structural genes of the anthocyanin biosynthetic pathway (Table 1, Figure 2D) [108]. In *C. annum*, differential expression of the *MybA* (containing the R3R3 domain) and *Myc* genes was correlated with the accumulation of anthocyanins in flowers and fruits but not in foliar tissues [109]. VIGS silencing of a *MYB* gene in *C. eximium* resulted in a decrease in the expression of chalcone synthase (*CHS*), chalcone isomerase (*CHI*), flavonoid 3′,5′-hydroxylase (*F3´5´H*), dihydroflavonol 4-reductase (*DFR*), and UDP-glucose: flavonoid 3-O-glucosyltransferase (*3GT*) genes, whereas silencing of *WD40* led to a significant reduction in *CHS*, *F3H*, *F3´5´H*, *DFR* and *3GT* transcription levels [48]. WD40 proteins are a family belonging to the β-propeller protein group and characterized by the presence of a core region composed of 40 amino acids flanked by a glycine-histidine dipeptide and a tryptophan-aspartate dipeptide (WD) [110]. *C. annuum* L. CaMYC (belonging to the bHLH family) has been described as an important regulator of anthocyanin biosynthesis by increasing the expression of structural genes, and together with CaMYB and CaWD40, CaMYC controls the transcription of genes encoding enzymes involved in the biosynthesis of anthocyanins in chili pepper [40].

## 6. Transcriptional Regulation of the Ascorbic Acid (AsA) Biosynthetic Pathway

At the mature stage, *Capsicum* fruits contain high levels of ascorbic acid (also denominated ascorbate or vitamin C) [111], which has antioxidant properties and constitutes an important micronutrient of chili pepper [9,11,112]. AsA has been cataloged as an uncommon antioxidant agent because it donates a single reducing equivalent, and monodehydroascorbate (the radical that AsA forms) reacts with radicals instead of with nonradicals [113]. In plants, ascorbate is involved in important metabolic processes, such as photosynthesis, photoprotection, and environmental stress and pathogen attack responses; additionally, AsA has a key role in signaling during cell division and cell expansion [114]. In humans, AsA plays an important role in health promotion because it has immunomodulatory and antimicrobial effects [115] and analgesic properties [116], and its effect has been recently evaluated for the treatment and prevention of SARS-CoV-2-related disease (COVID-19) [117]. Additionally, it plays an important role in the treatment and prevention of cancer [118].

The content of AsA varies among varieties of chili pepper fruits, with a gradual increase of ascorbic acid occurring based on the state of fruit development from green to red [119] and a subsequent decrease occurring at the late ripening stage (partially red and fully red) [120]. In sweet pepper, the AsA concentration increased by 47% during the transition from green to orange fruit, and it was correlated with the conversion of chloroplasts into chromoplast, the loss of chlorophyll and the accumulation of carotenoids [121]. In the whole fruits of the serrano cultivar ‘Tampiqueño 74’, the concentration of AsA increased from 20 DPA, showed a maximum peak at 40 DPA, and was maintained at 50 and 60 DPA [122]. In plants, AsA is biosynthesized via the D-mannose/L-galactose pathway [123], and the L-galactose pathway has been reported to represent the main mechanism for the production of vitamin C in the leaves and fruits of chili pepper [122]. Previous reports indicated that the transcript levels of genes encoding the key enzymes of the AsA biosynthetic pathway were not positively correlated with the AsA concentration in the pericarp of chili pepper and suggested that these biosynthetic genes were downregulated during fruit development; nevertheless, high levels of the transcript encoding the ascorbate oxidase (AO) enzyme were reported [121,122]. Previous research has associated the degradation of AsA content with an increase in ascorbate oxidase and ascorbate peroxidase (APX) activity [124,125]. These results suggest a feedback mechanism in the regulation of AsA in chili pepper fruits [121].

A transcriptional analysis conducted in varieties of chili pepper with contrasting levels of AsA revealed that variations in the concentrations of these metabolites are related to the differential expression of key genes involved in the biosynthesis, degradation and recycling of AsA [126]. Moreover, a recent report indicated that the transcript profiles of L-galactano-1,4-lactone dehydrogenase (*GLDH*) (encoding an important AsA biosynthetic enzyme) were positively correlated with the *CaMYB16* gene, suggesting that CaMYB16 could be a transcriptional regulator of the biosynthesis of AsA in chili pepper fruits (Table 1, Figure 2E) [47].

## 7. Conclusions

The abundance and diversity of bioactive compounds accumulated during different stages of development and ripening in non-climacteric fruits represent very interesting study models. Nevertheless, transcriptional regulation of the ripening process in non-climacteric fruits has been poorly understood until now. Transcriptional control of genes encoding enzymes that catalyze biosynthetic pathways during ripening has recently been studied in the case of capsaicinoid and anthocyanin production. R2R3 MYB TFs have been identified as the master regulators of capsaicinoid biosynthesis, and a MYB TF interacting with MYC and WD40 TFs has been shown to regulate anthocyanin biosynthesis in *Capsicum* spp.

Although carotenoids and ascorbic acid are important bioactive compounds with a diversity of nutraceutical applications, the mechanisms underlying the transcriptional regulation of chromoplast biogenesis and carotenoid biosynthesis in chili pepper fruits have not yet been established.

Comprehensive genomic, transcriptomic, and epigenomic analyses provide valuable information about the up- and down-regulation of genes encoding TFs and different structural biosynthetic enzymes. Correlation analyses between the transcriptional profiles of TFs and structural biosynthetic genes and the pattern of accumulation of metabolites across fruit development might be very useful for identifying and selecting candidates for functional gene evaluations. After selecting possible regulatory candidates and performing gene function studies by silencing or editing genes encoding TFs through VIGS and cluster regulatory interspaced palindromic repeats (CRISPR-Cas9), respectively [44,127], Y1H interaction assays could be used to establish the regulatory role of different TFs in the ripening process of chili pepper. Additionally, protein–protein interaction assays (Y2H experiments) should be performed in the future to elucidate the regulatory interactions between different proteins involved in the fruit ripening process. Moreover, ChIP-sequencing (ChIP-seq) has become an important tool for testing the interaction between chromatin and DNA, in order to identify TF targets, and to determinate how TFs could regulate signal networks that affect the plant phenotype [127]. Finally, a novel technic named reverse chromatin immunoprecipitation (R-ChIP) has allowed the characterization of chromatin composition and the isolation of upstream TFs that regulate a particular gene [128]. These experimental alternatives might corroborate the regulatory role of TF candidates on the expression of genes encoding biosynthetic enzymes of chili pepper fruits.

## Figures and Tables

**Figure 1 ijms-22-12151-f001:**
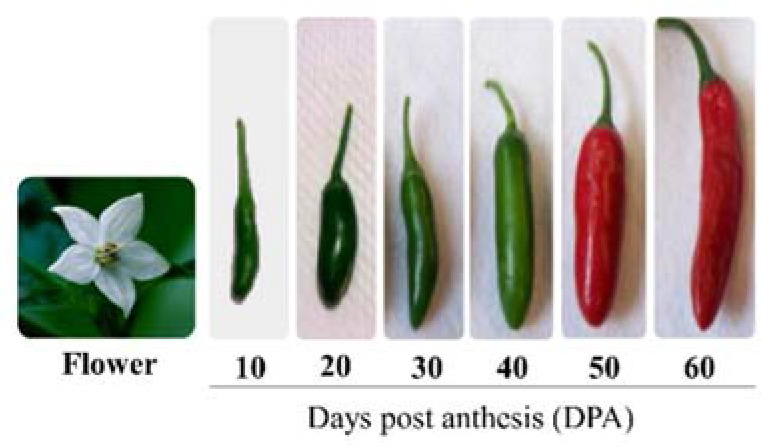
Ripening process of *Capsicum annuum* cv. ‘Tampiqueño 74’ fruits.

**Figure 2 ijms-22-12151-f002:**
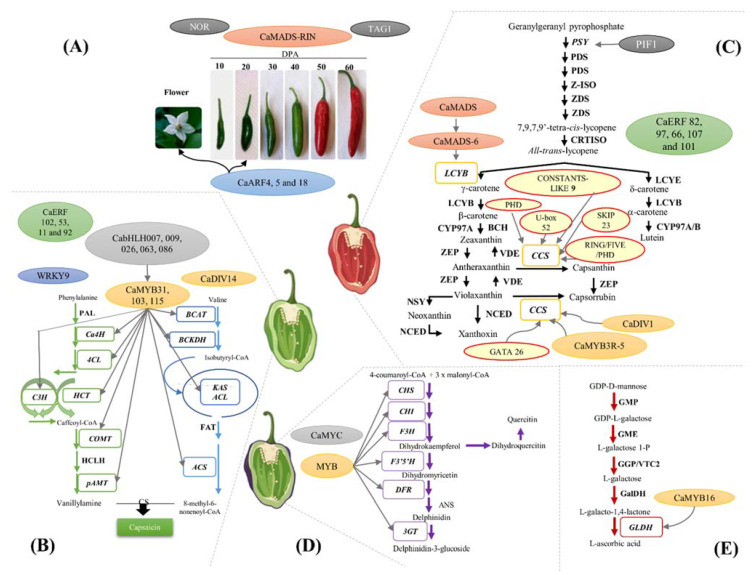
Transcription factors involved in the ripening process of *Capsicum* spp. (**A**) Chili pepper fruit development. Days post-anthesis (DPA). (**B**) Capsaicinoid biosynthetic pathway. Phenylalanine ammonia lyase (PAL), cinnamate 4-hydroxylase (Ca4H), 4-coumaroyl-CoAligase (4CL), hydroxycinnamoyl transferase (HCT), coumaroyl shikimate/quinate3-hydroxylase (C3H), caffeic acid *O*-methyltransferase (COMT), caffeoyl-CoA 3-*O*-methyltransferase (HCLH), aminotransferase (pAMT), branched-chain amino acid transferase (BCAT), isovalerate dehydrogenase (BCKDH), ketoacyl-ACPsynthase (KAS), acyl carrier protein (ACL), acyl-ACP thioesterase (FAT), acyl-CoA synthetase (ACS), and capsaicinoid synthase (CS). (**C**) Carotenoid biosynthetic pathway. Phytoene synthase (PSY), phytoene desaturase (PDS), ζ-carotene isomerase (Z-ISO), ζ-carotene desaturase (ZDS), carotene isomerase (CRTISO), β-lycopene cyclase (LCYB), ε-lycopene cyclase (LCYE), β-carotene hydroxylase (BCH), β-carotene hydroxylase cytochrome 450 type A and B (CYP), zeaxanthin epoxidase (ZEP), violaxanthin epoxidase (VDE), capsanthin-capsorubin synthase (CCS), neoxanthin synthase (NSY), and 9-*cis*-epoxycarotenoid dioxygenase (NCED). (**D**) Anthocyanin biosynthetic pathway. Chalcone synthase (CHS), chalcone isomerase (CHI), flavanone 3-hydroxylase (F3H), flavonoid 3′-hydroxylase (F3′H), flavonoid-3′,5′-hydroxylase (F3′5′H), dihydroflavonol 4-reductase (DFR), anthocyanidin synthase (ANS), and UFGT: UDP-Glc-flavonoid 3-O-glucosyl transferase (3GT). (**E**) L-ascorbic acid biosynthetic pathway. GDP-D-mannose pyrophosphorylase (GMP), GDP-D-mannose-3′,5′-epimerase (GME), GDP-L-galactose phosphorylase (GGP/VTC2), L-galactose-1-dehydrogenase (GalDH), and L-galactano-1,4-lactone dehydrogenase (GLDH).

**Table 1 ijms-22-12151-t001:** Transcription factors involved in the fruit ripening of *Capsicum* spp.

Transcription Factor	Consensus Motif	Gene	Function during Ripening	Reference
ARF (Auxin-response factor)	TGTCTC TGTCCC TGTCAC	*CaARF4*, *CaARF5*, *CaARF18*	Flower and fruit development	[38]
bHLH (basic Helix-Loop-Helix)	E-box CANNTG	*CabHLH007*, *CabHLH009*, *CabHLH026*, *CabHLH063 CabHLH086*	Capsaicinoid biosynthesis regulation	[39]
		*CabHLH009* *CabHLH032* *CabHLH048* *CabHLH095* *CabHLH100*	Carotenoid accumulation	
		*CaMYC*	Anthocyanin biosynthesis	[40]
ERF	GCC box TAAGAGCCGCC	*CaERF102*, *CaERF53*, *CaERF111*, *CaERF92*	Capsaicinoid biosynthesis	[41,42]
		*CaERF82*, *CaERF97, CaERF66*, *CaERF107*, *CaERF101*	Carotenoid biosynthesis	[42]
MADS box	CArG	*CaMADS* *CaMADS6* *CaMADS-RIN*	Ethylene signaling	[25,31,37]
MYB	TAACTAAC	*CaMYB31* *CaMYB48* *CaMYB108 CaMYB103* *CaMYB115* *CaDIV14*	Capsaicinoid biosynthesis and accumulation	[43,44,45,46,47]
		*CaMYB3R-5* *CaDIV1*	Carotenoid biosynthesis Putative regulator of the *CCS* gene	[47]
		*MYB*	Anthocyanin biosynthesis	[48]
		*CaMYB16*	Ascorbic acid biosynthesis	[47]
F-box protein SKIP23			Carotenoid biosynthesis. Putative regulators of the *CCS* gene	[49]
GATA transcription factor 26				
U-box domain-containing protein 52				
zinc finger family FYVE/PHD-type				
RING/FYVE/PHD-type				
CONSTANS-LIKE 9				

## Data Availability

Not applicable.

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
