# Peer review of "Transcriptional Regulation of Ripening in Chili Pepper Fruits (Capsicum spp.)"

_ijms, 2021, doi:10.3390/ijms222212151_

Round 1
Reviewer 1 Report
The manuscript is a well-organized review of transcriptional regulation during the ripening process of chili peppers. The review comprehensively describes the transcription factors involved in flower and fruit development, synthesis and accumulation of capsaicinoids, carotenoids and anthocyanins. The review contains a lot of information that will help researchers in this field. In "conclusion", future perspectives are drawn, but experiments using the following techniques are also useful and should be included if necessary. -Y1H is mentioned as an experiment to analyze the interaction between transcription factors and target genes, but chromatin immunoprecipitation is also a useful method to analyze the interaction in vivo. -Genetic approaches are also needed to elucidate the function of transcription factors and biosynthetic enzymes. It would be better to explain the genetic approach along with the current status of experimental systems for genetic modification and genome editing of chili peppers.Author Response
Reviewer 1
Comment about English language and style
(x) English language and style are fine/minor spell check required
Response: An English edition service (American Journal Experts) was used for the correction of this manuscript previously to the former submission, and the file of the edition certificate was attached in the original version.
Comments and Suggestions for Authors:
“The manuscript is a well-organized review of transcriptional regulation during the ripening process of chili peppers. The review comprehensively describes the transcription factors involved in flower and fruit development, synthesis and accumulation of capsaicinoids, carotenoids and anthocyanins. The review contains a lot of information that will help researchers in this field. In "conclusion", future perspectives are drawn, but experiments using the following techniques are also useful and should be included if necessary. -Y1H is mentioned as an experiment to analyze the interaction between transcription factors and target genes, but chromatin immunoprecipitation is also a useful method to analyze the interaction in vivo. -Genetic approaches are also needed to elucidate the function of transcription factors and biosynthetic enzymes. It would be better to explain the genetic approach along with the current status of experimental systems for genetic modification and genome editing of chili peppers”
Response: We really appreciate your comments to improve our manuscript. According to your suggestions, we modified the last paragraph of Conclusions (line 543-562), by incorporating other experimental approaches for the demonstration of the regulatory role of candidate transcription factors (formerly identified by co-expression analysis) on the expression of structural biosynthetic genes. We also added References 127 and 128.
Reviewer 2 Report
This is a review on transcriptional regulation of ripening in chili pepper fruit. The authors do a good job presenting the latest developments of the field. The manuscript is well-written and well-documented.
Author Response
Reviewer 2
Comment about English language and style
(x) English language and style are fine/minor spell check required
Response: An English edition service (American Journal Experts) was used for the correction of this manuscript previously to the former submission, and the file of the edition certificate was attached in the original version.
Comments and Suggestions for Authors:
This is a review on transcriptional regulation of ripening in chili pepper fruit. The authors do a good job presenting the latest developments of the field. The manuscript is well-written and well-documented.
Response:
We really appreciate your comments about our manuscript